# Nor1 and Mitophagy: An Insight into Sertoli Cell Function Regulating Spermatogenesis Using a Transgenic Rat Model

**DOI:** 10.3390/ijms26189209

**Published:** 2025-09-20

**Authors:** Bhola Shankar Pradhan, Deepyaman Das, Hironmoy Sarkar, Indrashis Bhattacharya, Neerja Wadhwa, Subeer S. Majumdar

**Affiliations:** 1Cellular Endocrinology Laboratory, National Institute of Immunology, Aruna Asaf Ali Marg, JNU Complex, New Delhi 110067, India; subeer@niab.org.in; 2Łukasiewicz Research Network, PORT Polish Center for Technology Development, 147 Stabłowicka Street, 54-066 Wrocław, Poland; 3Cell Biology and Bacteriology Laboratory, Department of Microbiology, Raiganj University, Raiganj 7337342, India; deepyaman.das@gmail.com (D.D.); h.sarkar@raiganjuniversity.ac.in (H.S.); 4Department of Zoology, Raiganj University, Raiganj 733134, India; 5Department of Zoology, Central University of Kerala, Periye Campus, Kasaragod 671316, India; indrashis.bhattacharya@cukerala.ac.in; 6Embryo Biotechnology Lab, National Institute of Immunology, New Delhi 110067, India; neerja@nii.ac.in; 7Gujarat Biotechnology University, GIFT City Campus, Gandhinagar 382355, India

**Keywords:** Sertoli cell, germ cell, mitochondria, *Nor1*, mitophagy, male fertility

## Abstract

Male infertility is a global health concern, and many cases are idiopathic in nature. The development and differentiation of germ cells (Gcs) are supported by Sertoli cells (Scs). Differentiated Scs support the development of Gcs into sperm, and hence, male fertility. We previously reported on a developmental switch in Scs around 12 days of age onwards in rats. During the process of the differentiation of Scs, the differential expression of mitophagy-related genes and its role in male fertility are poorly understood. To address this gap, we evaluated the microarray dataset GSE48795 to identify 12 mitophagy-related hub genes, including B-Cell Leukemia/Lymphoma 2 (*Bcl2*) and FBJ murine osteosarcoma viral oncogene homolog (*Fos*). We identify Neuron-derived orphan receptor 1 (*Nor1*) as a potential mitophagy-related gene of interest due to its strong regulatory association with two hub genes, *Bcl2* and *Fos*, which were differentially expressed during Sc maturation. To validate this finding, we generated a transgenic rat model with the Sc-specific knockdown of *Nor1* during puberty. A functional analysis showed impaired spermatogenesis with reduced fertility in these transgenic rats. Our findings suggest that *Nor1* may be an important mitophagy-related gene regulating the function of Scs and thereby regulating male fertility.

## 1. Introduction

The increase in cases of male infertility worldwide has become a global concern [1,2,3]. Recent reports suggest the prevalence of infertility is around 17.5% [4,5,6]. The causes of male infertility are diverse, and most of these cases are idiopathic in nature [7,8,9]. Moreover, most of these cases do not respond to the conventional modes of therapy, including hormonal supplementation [10,11,12,13]. Therefore, there is a need to explore the cellular and molecular mechanism underlying male infertility.

The development and function of Scs are important for the process of spermatogenesis, which regulates the division, differentiation, and survival of male Gcs, which develop into sperm [14,15]. The proper function of Scs is critical for male fertility [16]. Upon maturation, Scs support the robust process of spermatogenesis [17,18]. However, the maturation of Scs is impaired in many patients with idiopathic male infertility [19]. Thus, a better understanding of the molecular mechanisms regulating the process of Sc maturation may provide key insights into the etiology of male infertility.

One such pathway involves mitochondrial homeostasis and its dysfunction, which contribute significantly to the rising number of cases of male infertility [20,21,22]. Mitochondria are essential for energy production in all testicular cell types, including Scs [23,24]. The proper function of mitochondria is regulated by several quality control mechanisms such as mitochondrial fusion and fission, as well as mitophagy [25,26]. Mitophagy is the removal of excessive mitochondria or damaged mitochondria that are beyond repair [27,28,29,30]. During the process of mitophagy, the microtubule-associated protein 1A/1B light chain 3 (LC3) is recruited to the autophagosomal membrane, further binding to the damaged mitochondria expressing mitophagy receptors. Later on, mitophagosomes fuse with lysosomes for their degradation [31,32,33]. This process of the outer membrane fusion of mitochondria is mediated by mitofusins: mitofusin 1 (*Mfn1*) and mitofusin 2 (*Mfn2*), and the fusion of the inner membrane is mediated by Optic Atrophy 1, mitochondrial dynamin-like GTPase (*Opa1*). The dysfunction of *Mfn1*, *Mfn2*, or *Opa1* leads to various diseases due to compromised OXPHOS activity [34,35,36,37]. Moreover, the cascade of events involving phosphatase, tensin homolog (PTEN)-induced kinase I (PINK1), a serine/threonine kinase, and E3 ubiquitin ligase Parkin activate the ubiquitin proteasome system (UPS) [38,39] and recruit autophagosomes for mitophagy [40,41,42]. These pathways suggest that mitochondrial quality control may have an important role in the process of spermatogenesis.

Indeed, in many infertile male patients, various mutations in mitochondrial DNA (mtDNA) have been reported [43,44,45]. Mitochondrial dysfunction has been shown to be correlated with increased oxidative stress and decreased respiratory capacity in sperm, reflecting impaired energy production, which is important for sperm motility [46]. These mutations also regulate mitochondrial metabolism and human fertility [47]. A mutation in the mitochondrial polymerase gamma (POLG) also contributes to male infertility by impairing mtDNA replication [45,48]. A mouse model carrying mutations in the proofreading subunit of the mtDNA polymerase gamma (PolGAD257A) exhibited male infertility due to mitochondrial dysfunction [49,50].

Similarly, the inhibition of mitochondrial fusion by the deletion of *Mfn1*–*Mfn2* in the Gcs in a mouse model leads to azoospermia [51]. Moreover, the disruption of mitochondrial fission through mitochondrial fission factor (*Mff*) deficiency in a mouse model showed a reduced sperm count and subfertility [52]. These results suggest the importance of the mitochondrial fusion and fission processes in spermatogenesis. Recent reports suggest that autophagy also plays an important role in post-meiotic spermatids [53,54]. The deletion of autophagy related 7 (*Atg7*), an important autophagy gene in primordial germ cells, inhibits acrosome biogenesis [54]. This evidence further extends to Scs. Excess mitochondria agglomerate into residual bodies for their phagocytic degradation by Scs just before spermiation [55]. Some data indirectly suggests higher mitophagic activity in immature, proliferative Scs as compared to aged Scs [56]. A similar pattern is observed in neural stem cells, in which active mitophagy clears away defective mitochondria during the differentiation process [57]. Ethanol exposure has been shown to enhance mitophagy in rat Scs as a response to stress conditions, which is important for the survival of Scs [58]. Scs utilize autophagy and mitophagy for degrading ingested apoptotic substrates and maintaining the testicular microenvironment [59]. An important transcription factor GATA binding protein 4 (GATA4), which is required for the survival of Scs, is also regulated by mitophagy [60]. All these pieces of evidence suggest the dynamic nature of mitophagy may regulate the function of Scs. They also support the notion that there is a change in mitophagy-related activity during the differentiation of Scs. However, the specific roles of mitophagy-related genes (MRGs) in the development of Scs and male fertility are not fully understood. This gap in the literature forms the basis of our investigation. Therefore, we plan to identify and characterize the role of MRGs in the function of Scs and male fertility in this study.

Our previous work provides the foundation for this study. We reported that Scs remain in an immature, proliferative state in 5-day-old rats [61]. We also characterize the differential gene expression profiles of Scs across different developmental stages, i.e., immature (5-day-old rats), maturing (12-day-old rats), and mature (60-day-old rats) using the GEO dataset GSE48795 [62]. In the current study, we evaluated the microarray dataset GSE48795 to identify 12 mitophagy-related hub genes, including *Bcl2* and *Fos*. We identify *Nor1*, also known as Nuclear Receptor 4A3 (*NR4A3*)*,* as a potential MRG gene of interest due to its strong regulatory association with two hub genes, *Bcl2* and *Fos,* which were differentially expressed during Sc maturation. Importantly, *Nor1* was upregulated in maturing Scs (12-day-old rats).

*Nor1*, a pro-apoptotic factor, is a member of the NR4A subfamily [63]. Previous in vitro studies show the association of *Nor1* with mitophagy. For instance, NOR1 translocates to the mitochondria under proinflammatory conditions and promotes mitochondrial fragmentation via mitophagy in human beta cells [64]. Further, the knockdown of *Nor1* in C2C12 myotubes reduces peroxisome proliferator-activated receptor gamma coactivator 1α (*Pgc1α*), Transcription factor A, and mitochondrial (*Tfam*) and *Mfn2* levels, whereas the upregulation of *Nor1* in HeLa cells promotes apoptosis through a mitochondria-dependent pathway [65]. The loss of *Nor1* is embryonically lethal, and even a partial loss leads to lethality in some cases in mice [66]. On the other hand, a mutation of the *Nor1* gene locus leads to the generation of viable mice with only minor defects in inner ear development [67]. In mice, the Sc-specific knockdown of *Nor1* leads to compromised spermatogenesis due to enhanced levels of beta-catenin and SMAD family member 3 (*Smad3*) [68]. Based on this evidence, we generated a transgenic rat model with Sc-specific *Nor1* knockdown to test the importance of the MRG gene(s) associated with male fertility. Our findings revealed that *Nor1*-deficient rats were infertile, providing strong evidence suggesting the involvement of MRGs during Sc development and male fertility.

## 2. Results

### 2.1. Identification of Mitophagy-Related Genes (MRGs) During the Development of Scs

To investigate the role of MRGs in Sc development, we analyzed microarray data from the GSE48795 dataset to identify differentially expressed genes (DEGs). An outline of the bioinformatic workflow for screening hub genes is shown in Figure 1.

We used an adjusted *p* value of <0.05 and a cutoff of |log_2_(FC)| > 1 to identify significant DEGs. We compared the transcriptomic profile of a 5-day-old Sc with a 12-day-old Sc and identified a total of 1402 DEGs. Similarly, by comparing the RNA-seq data of a 5-day-old Sc with that of a 60-day-old Sc, we identified a total of 4612 DEGs. Then, we referred to a list of the mitophagy-related differentially expressed genes (MRDEGs) of humans from the PubMed website using the term “mitophagy-related genes” [69,70] and GeneCards database [71]. The GeneCards database provides comprehensive information about human genes. A total of 3073 MRGs were obtained after combining the results and removing duplicates. Further, we identified MRDEGs in rats. For this, we initially identified the orthologs for these MRDEGs of humans in rats from the NCBI gene database. After the successful identification of the orthologs, we screened 99 and 455 MRDEGs for rats in Sc development from 1402 (5- vs. 12-day-old Scs) and 4612 (5- vs. 60-day-old Scs) DEGs, respectively (Figure 2 and Appendix A).

Thus, there were a total of 488 mitophagy-related genes in Sc development (MRGSCD) considering both the phases. A total of 66 MRGSCDs were identified to be common between the two experimental conditions, i.e., 5d vs. 12d and 5d vs. 60d (Figure 2). Thus, these 66 genes can be considered to be crucial mitophagy-related genes. A functional enrichment analysis of the common 66 MRGSCDs showed that GO biological processes like GO:0022414—“reproductive process” and GO:0032502—“developmental process” were enriched (Figure 3). Similarly, GO molecular functions like GO:0005488—“binding” were enriched (Appendix A).

The reconstruction of a protein interaction network for 66 common MRGSCDs would unravel the vital signaling cascade of mitophagy-related genes in Sc development common to both stages. So next, with these 66 MRDEGs, we reconstructed a protein–protein interaction network (PPIN) using STRING v12 (with a confidence score > 0.4) (Figure 4).

This PPIN consists of 33 nodes and 76 edges. In a PPIN, hub genes are the nodes with the greatest number of edges or the most connections. Subsequently, using a network analyzer in Cytoscape 3.10.3., we identified 12 hub genes in the PPIN, viz. epidermal growth factor receptor (*Egfr*), *Bcl2*, C-C motif chemokine ligand 2 (*Ccl2*), matrix metallopeptidase 2 (*Mmp2*), insulin-like growth factor 1 (Igf1), fibroblast growth factor 7 (Fgf7), apolipoprotein E (*Apoe*), *Fos*, C-X-C motif chemokine ligand 12 (*Cxcl12*), Occludin (*Ocln*), decorin (*Dcn*), and synuclein alpha (*Snca*) (Figure 4). Notably, among these 66 common genes, *Nor1* was highlighted as a key gene of interest due to its significant upregulation in both 12-day- and 60-day-old Scs (Figure 2B and Appendix A). Moreover, it was found that *Nor1* was involved in the interactions with only two important hub genes, i.e., *Bcl2* and *Fos* in the PPIN. So, we performed further studies on *Nor1* to validate the findings of this analysis.

### 2.2. Validation of the Differential Expression of Nor1 in Scs

Since *Nor1* was identified as an important gene related to mitophagy in our analysis, we validated the expression of *Nor1* in the 5-day-old Scs and 12-day-old Scs, as it was upregulated in the microarray analysis (Figure 5A). These cells were treated with follicle-stimulating hormone (FSH) and testosterone (T), which were identical to those of the microarray samples. The q-RT-PCR data showed that *Nor1* was significantly (*p* ≤ 0.05) upregulated in the Sc isolated from the 12-day-old Scs as compared to that of the 5-day-old Scs (*n* = 4) (Figure 5B).

### 2.3. Generation of Transgenic Rat with Sc-Specific NOR1 Knockdown

Since we identified *Nor1* to be upregulated in a 12-day-old Sc from the rat microarray data, we generated a transgenic rat with the Sc cell-specific knockdown of *Nor1* (Tg rat) using the Pem promoter to determine the functional role of *Nor1* in spermatogenesis and mitophagy. As a control, we generated the transgenic rat with the Sc cell-specific knockdown of *LacZ* (LacZ control rat). The integration of the transgene was confirmed by slot blot analysis using the GFP probe (Figure 5C,D). We performed a q-RT-PCR analysis to determine the knockdown efficiency of *Nor1* in the testis of the Tg rat. We observed that the expression levels of *Nor1* in the Tg rat were reduced by 47% as compared to the LacZ control rat (Figure 5E). Since the Pem promoter drives the expression of shRNA against *Nor1* and *gfp*, we analyzed the testicular section of the Tg rats and age-matched wild-type (wt) control rats for the expression of GFP using anti-GFP antibodies to determine the transgene expression. We observed the expression of GFP in the testis of the Tg rat but not in that of the control (Figure 5F), suggesting that the Pem promoter was active in the Tg rats and able to drive the expression of shRNA.

### 2.4. Attenuation of Spermatogenesis in Testes of Sc-Specific Nor1 Knockdown Tg Rats

The Tg rat line could not propagate beyond the F1 generation as the positive male pups born from the mating of F1-generation siblings were infertile (Figure 6A). On the other hand, the LacZ control rats were maintained beyond the F1 generation. All of our studies were conducted using the F1 generation of shRNA knockdown rats, as the shRNA transgene behaves like dominant-negative alleles of the gene of interest; hence, Tg rats can be analyzed in the F1 generation itself without waiting for homozygosity.

The F1-generation male rats from the Tg rats and the LacZ control male rats were analyzed for defects in reproductive functions. The testis weight of the Tg rats (0.79 ± 0.11 g) decreased significantly (*p* ≤ 0.05) compared to that of the LacZ control rats (1.497 ± 0.02 g) (Figure 6B). Further, we analyzed the sperm count from the epididymis of the Tg rats and control. We observed a significant (*p* ≤ 0.05) reduction in sperm counts in the Tg rats (4 ± 1.52 × 10^6^/mL/epididymis) as compared to the age-matched LacZ control rat (144.7 ± 4.667 × 10^6^/mL/epididymis) (Figure 6C). Although the total sperm count was found to be drastically reduced, more than 80% of the sperm was found to be motile in the Tg rats.

Qualitative histopathological evaluations of the testes of the Tg rats (2 months old and 10 months old) and LacZ control rat (10 months old) were performed to determine the effect of NOR1 knockdown. We observed that that of the LacZ control rat showed a normal tubular structure, whereas those of the Tg rats showed the sloughing of spermatogenic cells in the majority of the tubules (Figure 6D). Moreover, multiple vacuoles were observed in the seminiferous epithelium of the Tg rat, suggesting that spermatogenesis was impaired due to the decline in *Nor1*.

## 3. Discussion

Our study provides novel insights into the role of MRGs during the development of Scs and their effect on spermatogenesis. We identified 12 hub MRGs (*Fos*, *Bcl2*, *Egfr*, *Ccl2*, *Mmp2*, *Igf1*, *Fgf7*, *Apoe*, *Cxcl12*, *Ocln*, *Dcn*, and *Snca*) that are differentially expressed during Sc differentiation. To assess their importance, we performed a functional genomics study and identified *Nor1* as a key MRG that interacts with two major hub genes, Bcl2 and Fos (Figure 2, Figure 3 and Figure 4). *Nor1* expression was significantly upregulated in 12-day-old Scs compared to 5-day-old Scs, and we validated its differential expression (Figure 5). To investigate the role of *Nor1* in mitophagy and spermatogenesis, we generated transgenic (Tg) rats with reduced *Nor1* expression specifically in Scs during puberty (Figure 5). Male Tg rats were infertile due to a low sperm count and displayed abnormalities in testicular architecture (Figure 6). Overall, our findings suggest that *Nor1* is an important MRG and plays a critical role in spermatogenesis in adult rats.

Several of the hub genes we identified show dynamic regulation during the development of Scs. *Bcl2*, a key member of the Bcl2 family of proteins, has an anti-apoptotic role [72]. Previous studies have shown that a reduction in levels of Bcl2 leads to apoptosis in various cell types [73]. Interestingly, *Bcl2*-deficient mice show normal spermatogenesis [74]. In infertile males, there are higher *Bcl2* mRNA levels, which indicate an attempt to prevent germ cell death [75]. It was upregulated in the 12 d old Scs in our analysis, suggesting that a higher level of *Bcl2* has a protective mechanism against cell death. Moreover, *Bcl2* is a known inhibitor of Parkin-mediated mitophagy, suggesting that it may regulate mitochondrial quality control in Scs [76].

*Fos*, primarily in the form of *cFos*, is a pivotal transcription factor. *cFos* upregulates Matrix metallo protease8 (*Mmp8*), which degrades gap junction proteins that play an important role in the development of Scs [77]. We observed a downregulation of *cFos* in 12 d old Scs onwards, which coincides with the upregulation of gap junction proteins during puberty. Further, elevated reactive oxygen species (ROS) levels can upregulate the level of *cFos* [78,79]. The stress-induced upregulation of *cFos* may be a protective mechanism for the cells [80].

During the metastasis of a cancer cell, it translocates to the mitochondria and regulates mitochondrial dynamics [81]. Interestingly, knockout mice models lacking EGFR ligands show normal spermatogenesis [82]. However, EGFR-overexpressing mice show infertility [83]. *Egfr* was observed to be downregulated during Sc differentiation, suggesting that its downregulation may be required for proper Sc maturation.

The chemokine *Ccl2* promotes autophagy and mitochondrial biogenesis [84,85]. *Ccl2* overexpression leads to mitochondrial accumulation [86]. We observed that its level was reduced in 12 d old Scs, suggesting that the downregulation of *Ccl2* may facilitate mitochondrial turnover during Sc maturation. Similarly, *Mmp2* modulates mitochondrial proteins [87] and its expression is regulated by FSH in Scs [88]. Its level was reduced in the 12 d old Scs, indicating that mitochondrial remodeling in Scs may be developmentally regulated.

*Igf1* is shown to activate mitophagy through nuclear respiratory factor 2/ Sirtuin 3 (Nrf2/Sirt3) signaling [89] and also regulates mitochondrial dynamics [90]. However, *Igf1* knockout mice are fertile [91]. Our data show that the Igf1 level was reduced during Sc maturation, suggesting that the downregulation may be a developmental shift in energy metabolism. We observed a similar pattern (downregulation in 12 d old Scs) for *Fgf7*, *Apoe*, *Fos*, *Cxcl12*, and *Dcn*, all of which are associated with mitochondrial or autophagy-related functions [92,93,94,95,96].

On the other hand, *Ocln* and *Snca* were upregulated in the differentiated 12 d old Scs. Importantly, Ocln knockout male mice are infertile [97]. It has a role in the structural integrity and mitochondrial stability of Scs [98]. The elevated level of *Ocln* during the development of Scs supports its crucial role in the function of Scs. The elevated level of *Snca* is associated with mitochondrial dysfunction [99] and excessive mitophagy [100]. These reports suggest that the regulation of *Scna* may have a role in mitochondrial homeostasis during Sc maturation.

The identification of these differentially expressed hub genes related to mitophagy suggests that developmental changes in Scs are associated with mitophagy. To further validate this, we selected *Nor1* for a detailed investigation. *Nor1* is also associated with two hub genes, *Bcl2* and *Fos*. Based on these findings, we generated a transgenic (Tg) rat with an Sc-specific reduction in *Nor1* to validate our analysis of mitophagy-related genes in Scs. We observed that Tg rats with Sc-specific NOR1 decline were infertile due to a reduced sperm count and abnormal testicular architecture (Figure 6). These findings are consistent with previous reports on *Nor1* [68], supporting the reliability of our in vivo functional analysis of *Nor1* and the identification of mitophagy-related hub genes.

Rat models were chosen because our microarray data are from rats, and rats are physiologically closer to humans than mice [101,102]. Mice are widely used for genetic studies due to the advancement in embryo transfer techniques. However, rats have distinct advantages for studying spermatogenesis. In particular, spermatogenesis requires approximately 55 days to complete a full cycle in a rat, which is closer to that of a human (64 days) [103,104,105]. Also, the rat seminiferous epithelium has 14 defined stages compared to 12 in mice [103,104].

The proximal Rhox-5 promoter (Pem promoter) drives the expression of target genes only in Scs from 14 days postnatally onwards [106]. We selected this promoter, as *Nor1* is upregulated from 12 days onwards in rats. On the other hand, the anti Mullerian hormone (*Amh*) promoter is active from birth to the neonatal stage. For instance, the expression of *Amh* in mouse Scs starts at embryonic day 12 dpc, which is gradually reduced during Sc differentiation and in adult testes [107]. The proximal promoter of *Amh* (180 bp) is sufficient for driving its expression specifically in Scs [108]. Therefore, we selected proximal Rhox-5 promoter (Pem promoter) to knockdown *Nor1* in Scs during puberty.

The limitation of this study is that we did not perform any studies on the functional aspect of mitophagy in these Tg rats due to the lower number of Tg rats generated by these lines. Further studies will be needed to determine the effect of *Nor1* on mitochondrial structure and function in these Tg rats.

In conclusion, our integrative genomic and functional analyses identified 12 key hub genes related to mitophagy that are differentially expressed during the development of Scs. Furthermore, we demonstrated that *Nor1* plays an important role in spermatogenesis in rat testes. These findings provide new insights into the roles of other MRGs during Sc development and further highlight the critical contribution of mitochondrial dysfunction to male infertility.

## 4. Materials and Methods

### 4.1. Animals

Wistar rats (*Rattus norvegicus*) were obtained from the Small Animal Facility of the National Institute of Immunology (New Delhi, India). The rats were housed in a climate-controlled facility under standard light conditions (14 h light/10 h dark cycle), temperature (23 °C), and humidity (50 ± 5%). The rats were housed in individually ventilated cages with (IVC) ad libitum access to acidified autoclaved water and were handled by trained personnel. Rats were used as per the guidelines laid down by the CPCSEA (Committee for the Purpose of Control and Supervision of the Experiments on Animals). Protocols for experiments were approved by the Institutional Animal Ethics Committee (IAEC) of National Institute of Immunology, India, constituted by CPCSEA (IAEC number: IAEC 220/09). Rats were euthanized by cervical dislocation as approved by the Institutional Animal Ethics Committee of National Institute of Immunology. For the primary Sc culture, about 100 (5-day-old male) rats and 60 (12-days-old male) rats were used. For the study involving transgenic rats, about 220 rats (including non-transgenic littermates) were used.

### 4.2. Screening of Hub Genes for Sertoli Cell Development in Rats

We identified differentially expressed genes (DEGs) by analyzing microarray data from the GSE48795 dataset [62] in order to examine the function of mitophagy-related genes in Sertoli cell development. The Bioconductor project’s basic packages in the NCBI Gene Expression Omnibus (GEO) database, which were based on limmav3.26.8, were utilized for data retrieval, logarithmic transformation, and quantile normalization of the data [109]. Benjamini–Hochberg (false discovery rate (FDR)) method was used for identifying significant DEGs. Considering this, an adjusted *p* value < 0.05 along with |log_2_(FC)| > 1 was considered to identify significant DEGs. A cut-off of adjusted *p* value < 0.05 and |log_2_(FC)| > 1 was considered to identify significant DEGs. The orthologs for humans in rats were identified from NCBI’s gene database [110]. Then, to reconstruct the PPIN for Sertoli cell development, we considered both physical and functional interactions from STRING v12 [111]. A default cut-off, i.e., confidence score > 0.4, was considered for the reconstruction of the PPIN. To identify hub genes, we used a network analyzer in Cytoscape 3.10.3 [112] to calculate network degree centrality. A network degree centrality > average of total degree count of the nodes, i.e., > 4.5, was considered the cut-off for identifying hub genes.

### 4.3. Functional Enrichment Analysis

We used DAVID v2023q4 for GO biological process and GO molecular function enrichment analyses of the genes [113].

### 4.4. Differential Expression Analysis of Nor1 in 5-Day-Old and 12-Day-Old Rat Sc Cultures

We identified *Nor1* as a candidate gene from the analysis. The differential expression of *Nor1* from this analysis was validated by q-RT-PCR analysis in the four separate sets of Sc cultures (*n* = 4 biological replicates) of 5-day- and 12-day-old rat testes. Testes from about 20–30 and 10–20 male rats were pooled for 5-day- and 12-day-old rat Sc cultures, respectively [61]. The Scs were isolated, cultured, and treated with FSH and testosterone in combination on day 4 of culture, as we reported previously [61,62]. Further, the total RNAs were extracted from these groups of cells as described previously [114,115]. Whole RNA was extracted using TRI reagent (Sigma Aldrich, St. Louis, MO, USA) as per the manufacturer’s instructions. The quantity and quality (260/280) of RNA were determined using NanoDrop 2000c spectrophotometer (Thermo Scientific, Waltham, MA, USA). The 260/280 and 260/230 absorbance ratios of all the RNA preparations used in this study were within the range of 1.8–2.0. This suggested that all the RNA preparations were of sufficiently good quality for gene expression analysis using real-time PCR. One microgram of RNA was treated with 0.5 U DNaseI (Thermo Scientific, Waltham, MA, USA) to remove any contaminating genomic DNA fragments. This was followed by single-strand c-DNA synthesis using M-MLV reverse transcriptase (Promega, Madison, WI, USA) as per the manufacturer’s protocol. qRT-PCR amplifications were performed using RealplexS (Eppendorf, Hamburg, Germany) in a total volume of 10 μL (1 μL of cDNA), with 0.5 μM of each primer and 5 μL of Power SYBR Green Master Mix (Applied Biosystems, Foster City, CA, USA).

### 4.5. Plasmids and Cloning

The knockdown of *Nor1* using shRNA in the Scs of the prepubertal rats was performed as described previously [68,116,117]. shRNA was designed by using online tools available at Dharmacon (https://rnaidesigner.thermofisher.com/rnaiexpress/setOption.do?designOption=shrna&pid=-38224801…; accessed on 17 May 2011) and Clontech (https://www.takarabio.com/assets/a/112504?srsltid=AfmBOopKDxPbPeL8erHa66yZk3C3LfUdHdyNvmriqXyHMvG-1…; accessed on 17 May 2011). shRNA sequences were verified for target specificity by BLAT (http://genome.ucsc.edu/cgi-bin/hgBlat?command=start) and BLAST 2.2.26 analysis (http://www.ebi.ac.uk/Tools/sss/ncbiblast/). The shRNAs against *Nor1* and bacterial *LacZ* (as a control) were designed and cloned under the Pem promoter. Pem promoter is reported to drive shRNA expression specifically in pubertal Sertoli cells [106]. The sequence for shRNA for *Nor1* was *TCGAGAGACAAGAGACGTCGAAATTTCAAGAGAATTTCGACGTCTCTTGTCTTTTTTTACCGGTCCGC,* and the sequence for shRNA for *LacZ* was *TCGAGGCATCGAGCTGGATAATAATTCAAGAGATTATTATCCAGCTCGATGCTTTTTTACCGGTCCGC*.

### 4.6. Generation of Transgenic (Tg) Rats with Reduced NOR1 in Scs

The construct was linearized with Stu I and was used to generate a transgenic rat (Tg) as we reported previously [101,114,118]. Briefly, the linearized construct (a total of 30 µg DNA with a concentration of 1 µg/µL) was injected into the testis of a 38-day-old Wistar rat and electroporated using 8 square 90 V electric pulses in alternating directions with a time constant of 0.05 second and an inter-pulse interval of ~1 second via an electric pulse generator. The electroporated male rats were mated with wild-type female rats 60 days post-electroporation. Pups that were born were screened by PCR using genomic DNA obtained from their tail biopsies, and those positive for the transgene were regarded as F1 generation of transgenic (Tg) rats.

The integration of transgene in the transgenic rats was confirmed by slot blot analysis as described previously [115,116]. For slot blot analysis, around 2 µg of gDNA isolated from the tail biopsies was denatured at 95 °C for 10 min and blotted onto Hybond N+ (Amersham Pharmacia Biotech, Buckinghamshire, UK) in a slot blot apparatus (Cleaver Scientific Co., Rugby, UK) under vacuum. The membrane was UV cross-linked at energy of 12 × 10^4^ µJ/cm^2^ in a CL-1000 Ultraviolet crosslinker (UVP, Upland, CA, USA). The non-radioactive DIG probe was used to detect the transgene. The probe was prepared by amplifying a fragment that contained part of the *gfp* (630 bp) using the following primers: GFPF: *GACGTAAACGGCCACAAGTT* and GFPR: *GGCGGTCACGAACTCCAG*. During PCR, the probe was labeled with DIG. The membranes were pre-hybridized for 2 h at 37 °C in hybridization solution without labeled probe and then hybridized separately at 37 °C with probe specific for GFP for 16 h. The membranes were washed twice, for 5 min each, at room temperature in 2× saline sodium citrate buffer and 0.1% SDS. This was followed by another washing for 15 min at 60 °C (in 0.1X saline sodium citrate buffer and 0.1% SDS). Detection was performed by using the DIG labeling and BCIP/NBT substrate according to the manufacturer’s recommendation (DIG system user’s guide for filter hybridization, Roche, Mannheim, Germany).

For this study, we generated two female founders for *Nor1* knockdown rats and two founders for *LacZ* knockdown rats. The efficiencies of generation of transgenic rats for both the lines were close to 33%, which is in agreement with our previous study, which showed that 33.08% of the progeny generated from single cohabitation of an electroporated fore founder (generated by us) with a wild-type female was transgenic [101].

### 4.7. Validation of Knockdown of NOR1 by q-RT-PCR

Total RNA was extracted from the whole testicular extract of control and 10-month-old Tg rats (*n* = 4 biological replicates) as we described previously [114,115]. q-RT-PCR amplifications were performed to detect expression of *Nor1* and *Ppia* (loading control) using the RealplexS (Eppendorf) as described previously [114,115]. Primers for *Nor1* were as follows: NF: *GCTTGTCTCTGCACCATTCA*; and NR: *TGAGCTAGGCCTCGAAGGTA*. Primers for *Ppia* were as follows: PF: *ATGGTCAACCCCACCGTGT*; and PR: *TCTGCTGTCTTTGGAACTTTGTCT*. The expressions of mRNA of the target genes were evaluated by the efficiency-corrected 2^−ΔΔCT^ method.

### 4.8. Immunofluorescence Microscopy

The testes of Tg rats (10 months old) (*n* = 3) and age-matched WT rats (*n* = 3) were collected and fixed in Bouins solution, and IF was performed as described previously [114,115]. To stain for transgene GFP, a primary rabbit polyclonal anti-GFP antibody was used at a dilution of 1:250 and incubated for 2 h at room temperature. A goat anti-rabbit IgG conjugated with Alexa 488 (Thermoscientific, Waltham, MA, USA) at a dilution of 1:1000 was used as the secondary antibody.

### 4.9. Tissue Histology

Tissue histology was performed as described previously [114,115]. Briefly, the testicular tissue samples of rats were fixed in Bouins solution at room temperature for 24 h. Dehydration of tissues was performed in a series of ascending concentrations of ethanol for 1 h at each grade of ethanol. The tissues were embedded in paraffin, and 4 µm sections were cut. Sections were stained with hematoxylin and eosin and were examined to evaluate the status of spermatogenesis in the control (10 months old) and Tg rats (2 months and 10 months old). Around 50 to 60 tubules per section were analyzed per rat.

### 4.10. Fertility Analyses of Nor1 Knockdown Rats

Testes weights of *Nor1* knockdown transgenic rats (10 months old) (*n* = 3) and *LacZ* shRNA knockdown control rats (*n* = 3) were recorded. The numbers of epididymal spermatozoa were analyzed as described previously [114,115]. Total numbers of sperm present in each epididymis were counted after releasing the sperm in 1 mL of 1× PBS by mincing the epididymis, and the average was determined. Motility of at least 200 epididymal spermatozoa per field was assessed under light microscope.

### 4.11. Statistical Analysis

For the validation of differential expression of *Nor1* by q-RT-PCR analysis, we used Mann–Whitney tests. For fertility studies, at least three or more sets of observations were analyzed by Mann–Whitney tests. A value of *p*  ≤  0.05 was considered significant. All statistical analyses were performed using GraphPad Prism v. 5.01 (GraphPad Software, La Jolla, CA, USA).

## Figures and Tables

**Figure 1 ijms-26-09209-f001:**
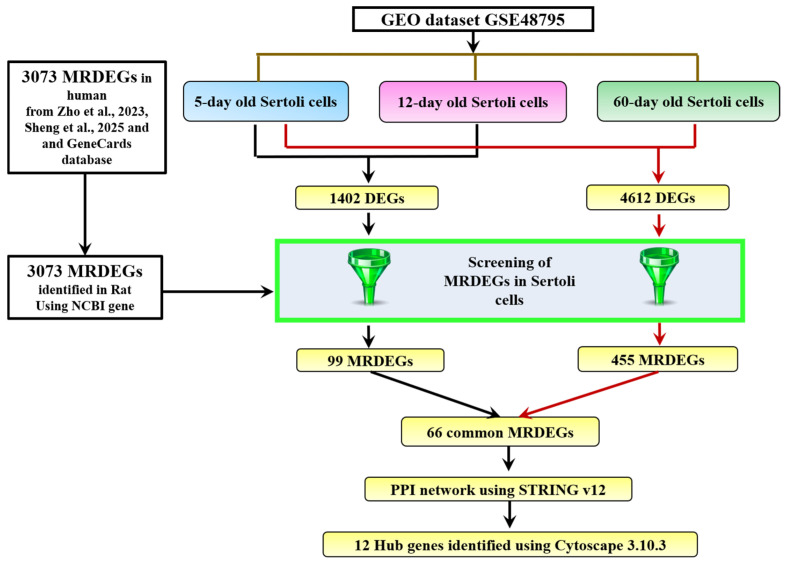
Workflow for screening mitophagy-related hub genes for Scs in rats. The MRDEG were selected as described in the text [69,70].

**Figure 2 ijms-26-09209-f002:**
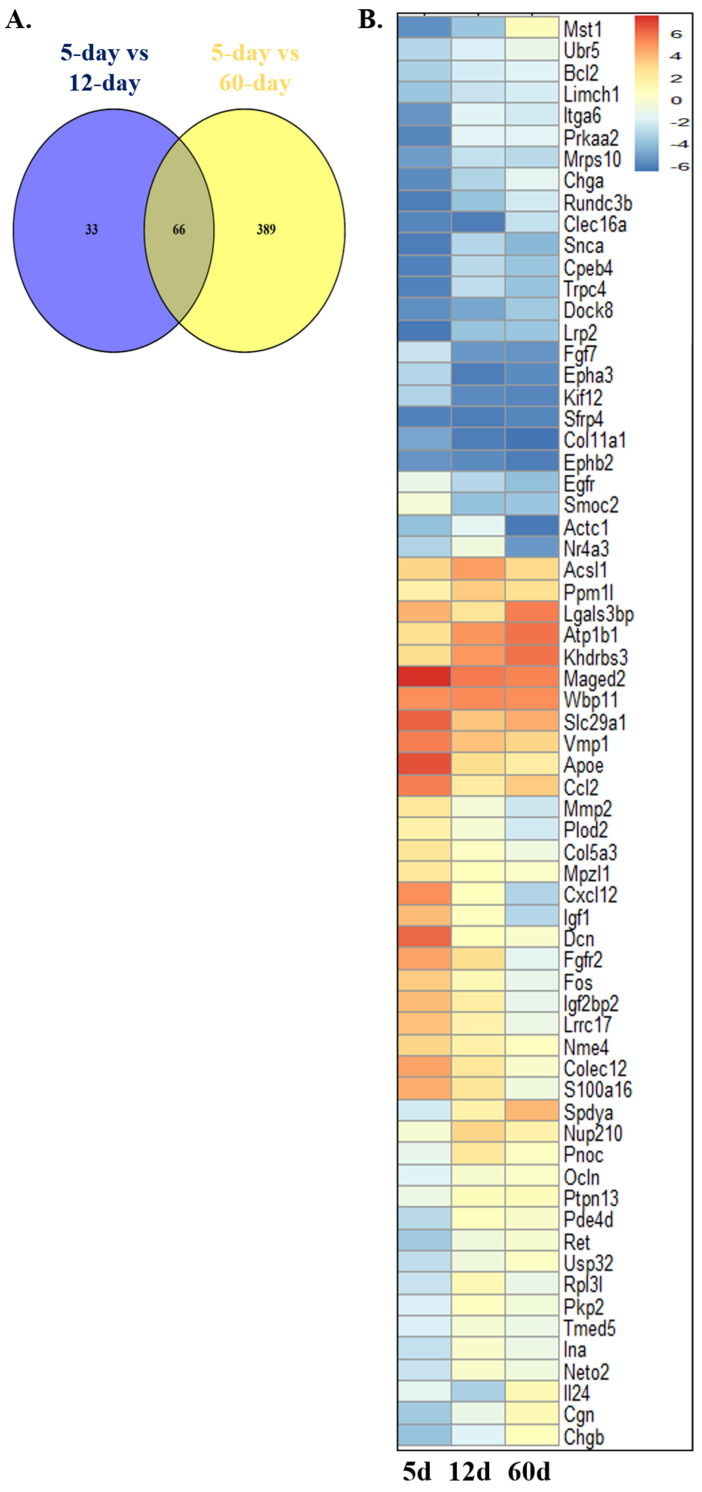
Identification of mitophagy related genes that are differentially expressed during Sc development in rats. (**A**) Venn diagram of common MRDEGs in the Scs of 5-day-, 12-day-, and 60-day-old rats. (**B**) Heatmap of average normalized expression of 66 common MRDEGs in the Scs of 5-day-, 12-day-, and 60-day-old rats. The red color denotes log_2_FC values > 1 and the blue color denotes log2FC values < 1.

**Figure 3 ijms-26-09209-f003:**
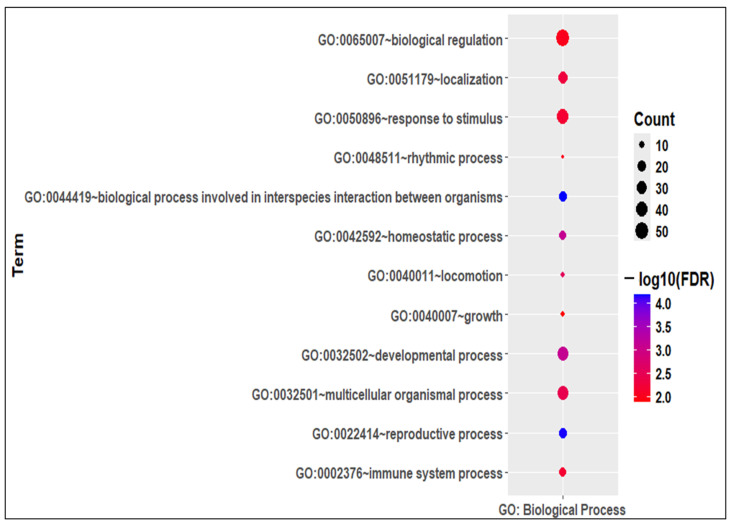
Functional enrichment of 66 common MRDEGs. GO biological process enrichment of 66 common MRDEGs in Scs.

**Figure 4 ijms-26-09209-f004:**
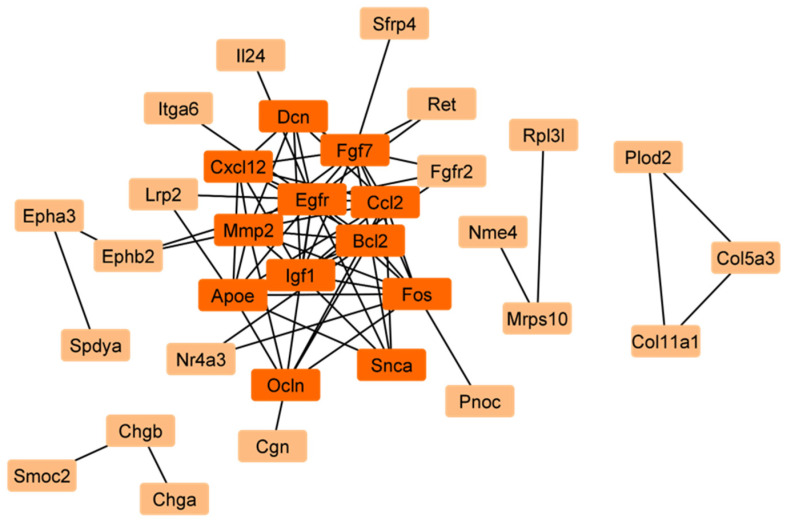
Protein–protein interaction network of mitophagy related to differentially expressed genes. Protein–protein interaction network of 66 mitophagy-related differentially expressed genes (common between Scs of 5- vs. 12- and 5- vs. 60-day-old rats) for Sertoli cell development in rats. This network consists of 33 nodes and 76 edges. The orange-colored nodes represent hub genes.

**Figure 5 ijms-26-09209-f005:**
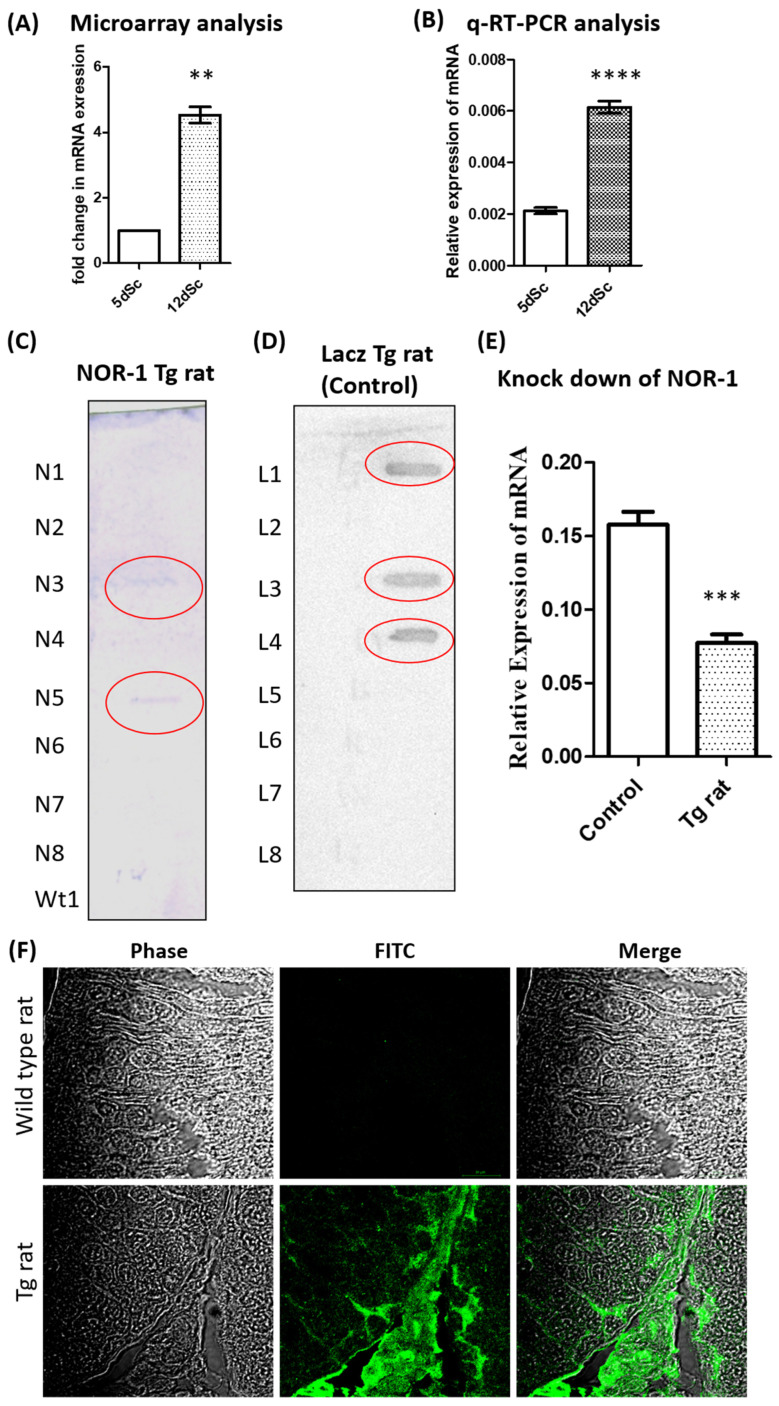
Upregulation of *Nor1* in the prepubertal Sc and generation of transgenic rat (Tg rat) with Sc-specific knockdown of *Nor1.* (**A**) *Nor1* was upregulated in the Sc of 12 d old rat as compared to the Sc of the 12 d old rat in the microarray analysis. *n* = 3; paired *t*-test; ** *p* ≤ 0.01. (**B**) Validation of *Nor1* upregulation in the Sc of 12 d old rat as compared to Sc of 5 d old rat by q-RT-PCR analysis. *Ppia* was used as an internal control. *n* = 4 (biological replicates). (**C**) The slot blot analysis of gDNA obtained from the tail of the progenies of the female founder Tg rat (with Sc-specific knockdown of *Nor1*). Progenies No 3 and No 5 were detected positive for transgenes (red circle). Wt denotes the gDNA obtained from the tail of the wild-type rat. N1 to N8 were the pup numbers obtained from the female founder. A part of the green fluorescent protein (GFP) was used as a probe to detect the integration of transgene. (**D**) The slot blot analysis of gDNA was obtained from the tail of the progenies of the control Tg rat (with Sc-specific knockdown of *LacZ*). L1 to L8 were the pup numbers obtained from the founder and the positive pups were annotated in red circle. (**E**) The expression of *NOR1* was significantly reduced in the testes of the Tg rats (10 months old) as compared to that of the control as detected by q-RT-PCR. *Ppia* was used as the internal control. *n* = 4 (biological replicates). (**F**) Detection of GFP in the testis of *Nor1* knockdown transgenic rat (Tg rat). The upper panel showed the expression of GFP was detected in the cross-section of the testis of a Tg rat (10 months old). GFP was used as a marker for the expression of transgene which included shRNA against *Nor1* driven by the Pem promoter. The lower panel showed the cross-section of the testis of the wild-type rats used as the control. *n* = 3. Mann–Whitney test. *** *p* ≤ 0.001; **** *p* ≤ 0.0001.

**Figure 6 ijms-26-09209-f006:**
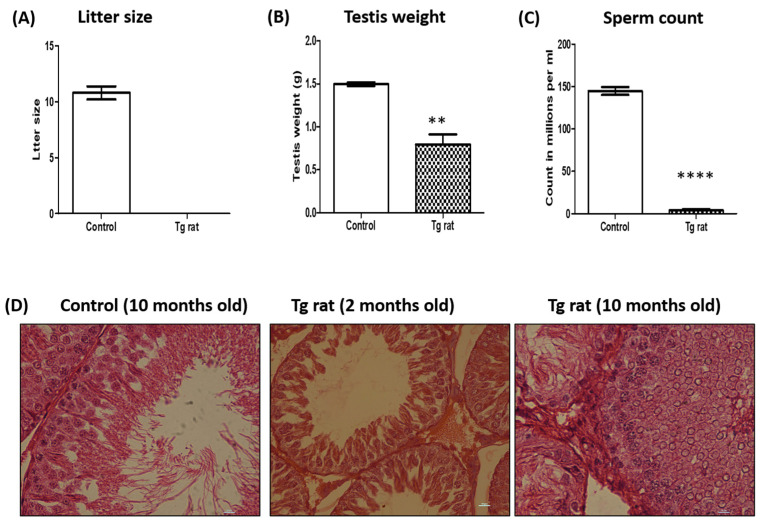
Fertility was inhibited due to impaired spermatogenesis in the Tg rats. (**A**) The litter size of the control and Tg rats. *n* = 3. (**B**) The testis weight of the Tg rats was reduced compared to that of the control. *n* = 3, Mann–Whitney test, ** *p* ≤ 0.01. (**C**) The sperm count of the Tg rats was reduced as compared to that of the control. *n* = 3, Mann–Whitney test, **** *p* ≤ 0.0001. (**D**) The hematoxylin and eosin staining of testicular sections obtained from the control rat (10 months old) and the Tg rats (2 months old and 10 months old). Spermatogenesis was inhibited in the Tg rats. The images were taken at 40× magnification. *n* = 3.

## Data Availability

The original contributions presented in this study are included in the article/Appendix A. Further inquiries can be directed to the corresponding author.

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
