# Peer review of "Nor1 and Mitophagy: An Insight into Sertoli Cell Function Regulating Spermatogenesis Using a Transgenic Rat Model"

_ijms, 2025, doi:10.3390/ijms26189209_

Round 1

Reviewer 1 Report

Comments and Suggestions for Authors

In this study, attempts were made elucidate the role of the differential expression (DE) of  myopathy-related hub genes in the testicular germ cells in the rats. This study provides information indicating that mitophagy-related genes could play important roles in the differentiation and maturation of rat testicular germ cells. The Reviewer suggests that the following comments would be helpful to improve the quality of the manuscript.

Comments:

1. What are the objectives of the study? The objectives need to be precisely given in this study.

2. It is difficult to follow the descriptions of the results because of inconsistencies with their presentations in the figures, poor visualization (Figures 1-2), inaccurate numbering of figures, poor explanations the cations, etc.

3. The two main genes of interest, Bcl2 (L276-281) and FOS (L312-315), were upregulated and downregulated in the Sertoli cells, respectively. Should consider to give more information about the role of above-mentioned genes in the differentiation and maturation of testicular germ cells. How do changes in their differential expression affect the functions of Sertoli cells?

4. Give the full names for all abbreviations/synonyms when used for the first time in the text (Introduction …..).

5. M & M

a) Give the number of rats used in this study.

b) More information about the RNA extraction procedure and quality control should be given (L442-443).

Comments on the Quality of English Language

The manuscript should be considered for publication after my comments have been completely addressed.

Author Response

We thank the reviewer for the constructive comments. We appreciate their critical feedback.  Based on the feedback of the reviewer, we have now revised the manuscript. Please find the point-by-point response for the reviewer’s comments below.

Comments and Suggestions for Authors

In this study, attempts were made elucidate the role of the differential expression (DE) of  myopathy-related hub genes in the testicular germ cells in the rats. This study provides information indicating that mitophagy-related genes could play important roles in the differentiation and maturation of rat testicular germ cells. The Reviewer suggests that the following comments would be helpful to improve the quality of the manuscript.

Response: We thank the reviewer for the critical reading of our manuscript.

Comments:

  1. What are the objectives of the study? The objectives need to be precisely given in this study.

Response: Uncovering mitophagy-related hub genes provides critical insight into Sertoli cell maturation process. Our identification of Nor1, connected to hub genes Bcl2 and Fos, and its functional validation in a transgenic rat model, demonstrates how mitochondrial quality control in Sertoli cells directly governs spermatogenesis and male fertility. These findings offer a mechanistic basis for infertility linked to defective mitophagy pathways.

We have now these lines L96 -99 in the manuscript.

However, the specific roles of mitophagy-related genes (MRG) in the development of Sc and on male fertility isnot fully understood. This gap in literature forms the basis of our investigation. Therefore, we plan to identify and characterize the role of MRG in the function of Sc and male fertility in this study.

  1. It is difficult to follow the descriptions of the results because of inconsistencies with their presentations in the figures, poor visualization (Figures 1-2), inaccurate numbering of figures, poor explanations the cations, etc.

Response: We understand that Figures 1 and 2 aren’t discernable due to lot of information in a small space. Now we have revised these two figures. The revised Figure 2 and Figure 3 and Supplementary Figure 1 are now added to the revised manuscript.  We hope this will improve the clarity of the data. We have also corrected the numbering of Figures and improved the captions of these figures.

  1. The two main genes of interest, Bcl2(L276-281) and FOS (L312-315), were upregulated and downregulated in the Sertoli cells, respectively. Should consider to give more information about the role of above-mentioned genes in the differentiation and maturation of testicular germ cells. How do changes in their differential expression affect the functions of Sertoli cells?

Response: We thank the reviewer for this suggestion. We have now discussed the role of these two genes in the context of Sertoli cell and mitochondria in the lines L273-278.

Several of the hub genes we identified show dynamic regulation during the development of Sc. Bcl2, a key member of the Bcl2 family of proteins has anti-apoptotic role [72]. Previous studies have shown that a reduction in level of Bcl2 lead to apoptosis in various cell types [73]. Interestingly, the Bcl2-deficient mice show normal spermatogenesis [74]. In the infertile males there is a higher Bcl2 mRNA levels, which indicates an attempt to prevent germ cell death [75]. It was upregulated in 12d Sc in our analysis suggesting that higher level of Bcl2 has a protective mechanism against cell death. Moreover, Bcl2 is a known inhibitor of Parkin-mediated mitophagy suggesting that it may regulate the mito-chondrial quality control in Sc [76].

Fos, primarily in the form of cFos, is a pivotal transcription factor. cFOS upregulate Matrix metallo protease8 (Mmp8) which degrade gap junction proteins that play im-portant role in the development of Sc [77]. We observed a downregulation of cFOS in 12d Sc onwards which is coinciding with the upregulation of gap junction proteins during puberty. Further, elevated reactive oxygen species (ROS) levels can upregulate the level of cFos [78, 79]. These stress induced upregulation of cFos may be a protective mechanism for the cells [80].

References

  1. Luo H, Peng F, Weng B, Tang X, Chen Y, Yang A, et al. miR-222 Suppresses Immature Porcine Sertoli Cell Growth by Targeting t he GRB10 Gene Through Inactivating the PI3K/AKT Signaling Pathway. Front Genet. 2020;11(581593).
  2. Gao Y, Wu H, He D, Hu X, Li Y. Downregulation of BCL11A by siRNA induces apoptosis in B lymphoma cell lines. Biomed Rep. 2013;Jan;1(1):47-52.
  3. Russell LD, Chiarini-Garcia H, Korsmeyer SJ, Knudson CM. Bax-dependent spermatogonia apoptosis is required for testicular devel opment and spermatogenesis. Biol Reprod. 2002;Apr;66(4):950-8.
  4. Steger K, Wilhelm J, Konrad L, Stalf T, Greb R, Diemer T, et al. Both protamine-1 to protamine-2 mRNA ratio and Bcl2 mRNA content in te sticular spermatids and ejaculated spermatozoa discriminate between fe rtile and infertile men. Hum Reprod. 2008;Jan;23(1):11-6.
  5. Hollville E, Carroll RG, Cullen SP, Martin SJ. Bcl-2 family proteins participate in mitochondrial quality control by regulating Parkin/PINK1-dependent mitophagy. Mol Cell. 2014;7;55(3):451-66.
  6. Chen Y, Wang J, Pan C, Li D, Han X. Microcystin-leucine-arginine causes blood-testis barrier disruption an d degradation of occludin mediated by matrix metalloproteinase-8. Cell Mol Life Sci. 2018;Mar;75(6):1117-1132.
  7. Lv D, Ji Y, Zhang Q, Shi Z, Chen T, Zhang C, et al. Mailuoshutong pill for varicocele-associated male infertility-Phytoche mical characterisation and multitarget mechanism. Front Pharmacol. 2022;13(961011).
  8. Nakamura BN, Lawson G, Chan JY, Banuelos J, Cortés MM, Hoang YD, et al. Knockout of the transcription factor NRF2 disrupts spermatogenesis in an age-dependent manner. Free Radic Biol Med. 2010;15;49(9):1368-79.
  9. Kyrgiafini MA, Sarafidou T, Mamuris Z. The Role of Long Noncoding RNAs on Male Infertility: A Systematic Review and In Silico Analysis. Biology (Basel). 2022;11(10).

  1. Give the full names for all abbreviations/synonyms when used for the first time in the text (Introduction …..).

Response: We have now added the abbreviations in the text and added a separate section in the manuscript in the line L508-550.

  1. M & M
  2. a) Give the number of rats used in this study.

Response: We have provided the number of rats used in this study in the lines L359-361.

For the primary Sc culture, about 100 (5 days old male rat) and 60 (12 days old male rat) rats were used. For the study involving transgenic rat, about 220 rats (including non-transgenic littermates) were used.

  1. b) More information about the RNA extraction procedure and quality control should be given (L442-443).

Response: We thank the reviewer for this suggestion. We have now added these informations in the lines L388 -400.

Whole RNA was extracted using TRI reagent (Sigma Aldrich) as per the manufacturer’s instructions. The quantity and quality (260/280) of RNA was determined using NanoDrop 2000c spectrophotometer (Thermo Scientific, Waltham, MA, USA). The 260/280 and 260/230 absorbance ratio of all the RNA preparations used in this study was within the range of 1.8 – 2.0. This suggested that all the RNA preparations were of sufficiently good quality for gene expression analysis using real-time PCR. One microgram of RNA was treated with 0.5 U DNaseI (Thermo Scientific, Waltham, MA, USA) to remove any contaminating genomic DNA fragments. This was followed by single-strand c-DNA synthesis using M-MLV reverse transcriptase (Promega, Madison, WI, USA) as per the manufacturer’s protocol. qRT-PCR amplifications were performed using the RealplexS (Eppendorf, Germany) in a total volume of 10 μl (1 μl of cDNA), 0.5 μM of each primer, and 5 μl of Power SYBR Green Master Mix (Applied Biosystems, CA, USA).

Comments on the Quality of English Language

The manuscript should be considered for publication after my comments have been completely addressed.

Response: We have now improved the quality of English in the manuscript.

Reviewer 2 Report

Comments and Suggestions for Authors

To the author

The study needs revision. All the sections should be revised and written in the scientific format

Non-specific comments

- Cross-check the references in the text and the full reference

- Cite the figures and tables in the results and discussion

- Check the expression, the tenses are not used correctly

- What is the basis of this study

- What is the significance of the study?

- Add more references to the background and the recent one.

-Mention the sample size in validation experiments briefly, since reviewers often look for that.

-Provide replication details (biological vs. technical).

- Include a schematic of bioinformatics and experimental workflow for clarity.

Title & Abstract

  • The title is clear and relevant, but you may want to explicitly add “transgenic rat model” to highlight the novelty.
  • In the abstract, clarify how the transcriptomic analysis led to focusing on NOR1. Right now, the transition from identifying hub genes → choosing NOR1 feels abrupt.

Introduction

  • Nicely builds rationale, but could be streamlined: there’s some repetition between the discussion of mitophagy in infertility and Sertoli cell development. Consider elaborating.
  • Consider sharpening the gap statement: “Although mitophagy has been implicated in germ cell biology, its role in Sertoli cell differentiation and spermatogenesis remains unclear

Materials and methods

  1. Animals

You describe housing and ethical approval well. Consider adding the number of animals used per group (5d, 12d, 60d rats; transgenic vs controls). This is critical for reproducibility.

  1. Screening of hub genes

The bioinformatics workflow is generally clear, but could benefit from a schematic (dataset → DEG identification → ortholog mapping → STRING/Cytoscape → hub genes). Specify which multiple-testing correction method was applied for adjusted p-values (Benjamini–Hochberg?).

  1. Differential expression analysis

You validated NOR1 expression with qRT-PCR; clearly state how many biological replicates were used (n=3 or n=4), and whether technical replicates were averaged.

  1. Plasmids and Cloning

Sequences are given, which is good. Add information on whether shRNA constructs were tested in vitro before moving to animal models.

  1. Transgenic Rat Generation

The electroporation method is well described, but clarify the efficiency (number of founders generated vs. screened).

Since the Pem promoter is Sertoli-specific, a short justification of why this promoter was chosen (vs. Amh or others) would strengthen the rationale.

  1. Validation of Knockdown

Good inclusion of qRT-PCR. Can you provide a CT validation curve?

Histology & Fertility Tests

For histology, state the number of tubules/sections analyzed per rat.

For sperm counts, indicate whether counts were averaged from both epididymides.

  1. Statistics

You mention Mann–Whitney tests, but for multiple comparisons (e.g., 5d vs 12d vs 60d), a Kruskal Wallis or ANOVA might be more appropriate. Clarify why non-parametric tests were chosen was normality tested?

Discussion

-Strong integration of findings with previous literature.

-Could benefit from highlighting species differences (rat vs. mouse) more explicitly, since you justify using rat models.

Author Response

We thank the reviewer for the constructive comments. We appreciate their critical feedback.  Based on the feedback of the reviewer, we have now revised the manuscript. Please find the point-by-point response for the reviewer’s comments below.

Reviewer

Comments and Suggestions for Authors

To the author

The study needs revision. All the sections should be revised and written in the scientific format

Response: We thank the reviewer for the suggestion. We have revised all the sections of the manuscript.

Non-specific comments

- Cross-check the references in the text and the full reference

Response: We have now updated the references.

- Cite the figures and tables in the results and discussion

Response: We have now corrected the number of the figures and updated their citations in the text.

- Check the expression, the tenses are not used correctly

Response: We have now corrected these mistakes in the text.

- What is the basis of this study

Response: Uncovering mitophagy-related hub genes provides critical insight into Sertoli cell maturation process. Our identification of Nor1, connected to hub genes Bcl2 and Fos, and its functional validation in a transgenic rat model, demonstrates how mitochondrial quality control in Sertoli cells directly governs spermatogenesis and male fertility. These findings offer a mechanistic basis for infertility linked to defective mitophagy pathways.

We have now these lines L96 -99 in the manuscript.

However, the specific roles of mitophagy-related genes (MRG) in the development of Sc and on male fertility isnot fully understood. This gap in literature forms the basis of our investigation. Therefore, we plan to identify and characterize the role of MRG in the function of Sc and male fertility in this study.

- What is the significance of the study?

Response: Male infertility remains a major global health problem, with many cases lacking a defined molecular cause. Sertoli cells are indispensable for germ cell development, and their maturation depends on mitochondrial integrity and quality control. This study demonstrates that mitophagy-related hub genes are one of the central regulators of Sertoli cell differentiation. Among them, Nor1, identified through its association with hub genes Bcl2 and Fos, is shown to be essential for spermatogenesis and male fertility using a transgenic rat model. These findings establish a mechanistic link between mitochondrial quality control and male reproductive capacity, providing novel molecular insights and potential therapeutic targets for idiopathic male infertility.

- Add more references to the background and the recent one.

Response: We have now added extra 30 relevant references (from the last five years) in the introduction and discussion section.

-Mention the sample size in validation experiments briefly, since reviewers often look for that.

Response: We thank the reviewer for this suggestion. We have now mentioned the sample size in the legends to the figures and also in the method section for all the experiments involving animal studies (Figures 5 and Figure 6).

Lines L359 -361

For the primary Sc culture, about 100 (5 days old male rat) and 60 (12 days old male rat) rats were used. For the study involving transgenic rat, about 220 rats (including non-transgenic littermates) were used.

-Provide replication details (biological vs. technical).

Response: We have now provided the biological replicates details in the legend to the figures (Figure 5 and Figure 6).

- Include a schematic of bioinformatics and experimental workflow for clarity.

Response: We thank the reviewer for this suggestion. We have now added the schematic as Figure 1 in the text in Line L130.

Figure 1. Workflow for screening of mitophagy related hub genes for Sc in rat

Title & Abstract

  • The title is clear and relevant, but you may want to explicitly add “transgenic rat model” to highlight the novelty.

Response: We thank the reviewer for this suggestion. We have modified the title as follows

NOR1 and Mitophagy: Insights into Sertoli Cell Function and Spermatogenesis Using a Transgenic Rat Model

  • In the abstract, clarify how the transcriptomic analysis led to focusing on NOR1. Right now, the transition from identifying hub genes → choosing NOR1 feels abrupt.

Response: We thank the reviewer for this suggestion. The modified abstract is provided below (in the line L18-33)

Male infertility is a global health concern and many cases of them are idiopathic in nature. The development and differentiation of germ cell (Gc) is supported by the Sertoli cell (Sc). The differentiated Sc supports the development of Gc into sperm, and hence male fertility. We previously reported that a developmental switch in Sc around 12 days of age onwards in rats. During the process of differentiation of Sc, the differential expression of mitophagy related genes and its role in male fertility is poorly understood. To address this gap, we evaluated the microarray dataset GSE48795 to identify 12 mitophagy-related hub genes, including B-Cell Leukemia/Lymphoma 2 (Bcl2) and FBJ murine osteosarcoma viral onco-gene homolog (Fos). We identify Neuron-derived orphan receptor 1 (Nor1) as a potential mitophagy-related gene of interest due to its strong regulatory association with two hub genes Bcl2 and Fos that were differentially expressed during Sc maturation. To validate this finding, we generated a transgenic rat model with Sc-specific knockdown of Nor1 during puberty. Functional analysis showed the impaired spermatogenesis with reduced fertility in these transgenic rats. Our findings suggest that Nor1 may be an important mitophagy-related genes regulating the function of Sc and thereby, regulate male fertility.

Introduction

  • Nicely builds rationale, but could be streamlined: there’s some repetition between the discussion of mitophagy in infertility and Sertoli cell development. Consider elaborating.

Response: We thank the reviewer for this suggestion. We have now elaborated these sections in the lines L68 -124.

  • Consider sharpening the gap statement: “Although mitophagy has been implicated in germ cell biology, its role in Sertoli cell differentiation and spermatogenesis remains unclear

Response: We thank the reviewer for this suggestion. We have now elaborated these sections in the lines L86 -99.

 Materials and methods

  1. Animals

You describe housing and ethical approval well. Consider adding the number of animals used per group (5d, 12d, 60d rats; transgenic vs controls). This is critical for reproducibility.

Response: We thank the reviewer for this suggestion. We have now mentioned the sample size in the legends to the figures and also in the method section for all the experiments involving animal studies (Figures 5 and Figure 6).

Lines L359 -361

For the primary Sc culture, about 100 (5 days old male rat) and 60 (12 days old male rat) rats were used. For the study involving transgenic rat, about 220 rats (including non-transgenic littermates) were used.

  1. Screening of hub genes

The bioinformatics workflow is generally clear, but could benefit from a schematic (dataset → DEG identification → ortholog mapping → STRING/Cytoscape → hub genes). Specify which multiple-testing correction method was applied for adjusted p-values (Benjamini–Hochberg?).

Response: We thank the reviewer for this suggestion. We have now provided this information in the lines L368 – 370.

Benjamini-Hochberg (False Discovery Rate (FDR)) method was used for identifying significant DEGs. Considering this, an adjusted P value < 0.05 along with |log2(FC)| > 1 was considered to identify significant DEGs.

  1. Differential expression analysis

You validated NOR1 expression with qRT-PCR; clearly state how many biological replicates were used (n=3 or n=4), and whether technical replicates were averaged.

Response: We thank the reviewer for this suggestion. n=4 biological replicates and the technical replicates were averaged. We have now added these information to the legend to the figure 5.

  1. Plasmids and Cloning

Sequences are given, which is good. Add information on whether shRNA constructs were tested in vitro before moving to animal models.

Response: We thank the reviewer for critical analysis. The primary Sertoli cell cultures are difficult to transfect. Therefore, we couldnot perform these in vitro experiments with the shRNA construct. The validation experiment (target and off target) were performed using bioinformatics analysis and it was mentioned in the text. However, our lab has vast experience in generating transgenic rat and transgenic mice using shRNA construct to knock down a particular gene in Sertoli cell in vivo (PMID: 23667645; PMID: 29064078; PMID: 28065881; PMID: 29363789; PMID: 29032151;  PMID: 33145986).

  1. Transgenic Rat Generation

The electroporation method is well described, but clarify the efficiency (number of founders generated vs. screened).

Response: We thank the reviewer for this suggestion. For this study, we generated two female founders for NOR1 knock down rats and two founders for LacZ knock down rats. The efficiency of generation of transgenic rat for both the lines were close to 33% and it is in agreement with our previous study which showed 33.08% of the progeny generated from single cohabitation of an electroporated fore founder (generated by us) with a wild-type female was transgenic (PMID: 27111419). These lines are now added to the text L442 -446.

Since the Pem promoter is Sertoli-specific, a short justification of why this promoter was chosen (vs. Amh or others) would strengthen the rationale.

Response: We thank the reviewer for this suggestion. We have now added these lines in the text L329-336

The proximal Rhox-5 promoter (Pem promoter) drives the expression of target gene only in Sc from 14 days postnatal age onward [106]. we selected this promoter as the Nor1 is upregulated 12 day onwards in rat. On the other hand, anti mullerian hormone (Amh) promoter is active from birth to neonatal stage. For instance, the expression of Amh in mice Sc starts at embryonic day 12dpc, gradually reduced during Sc differentiation and in adult testes [107]. The proximal promoter of Amh (180bp) is sufficient for drive its expression specifically in Sc [108]. Therefore, we selected proximal Rhox-5 promoter (Pem promoter) to knock down Nor1 in the Sc during puberty.

References

  1. Rao MK, Wayne CM, Meistrich ML, Wilkinson MF. Pem homeobox gene promoter sequences that direct transcription in a Se rtoli cell-specific, stage-specific, and androgen-dependent manner in the testis in vivo. Mol Endocrinol. 2003;Feb;17(2):223-33.
  2. Münsterberg A, Lovell-Badge R. Expression of the mouse anti-müllerian hormone gene suggests a role in both male and female sexual differentiation. Development. 1991;Oct;113(2):613-24.
  3. Giuili G, Shen WH, Ingraham HA. The nuclear receptor SF-1 mediates sexually dimorphic expression of Mu llerian Inhibiting Substance, in vivo. Development. 1997;May;124(9):1799-807.

  1. Validation of Knockdown

Good inclusion of qRT-PCR. Can you provide a CT validation curve?

Response: We thank the reviewer for critical analysis. The primer efficiency for Ppia was 0.83 based on our previous study (PMID: 22850685). Using the standard approach, we designed the primers for Nor1 as we did for Ppia. We did not perform the serial dilution experiment to determine the primer efficiency of Nor1 as we considered the primer efficiency to be 100±10 which is close to that of Ppia. In this study, our aim was to determine the relative expression level Nor1 between the two ages group and therefore, we did not perform the copy number analysis.  

Histology & Fertility Tests

For histology, state the number of tubules/sections analyzed per rat.

Response: We thank the reviewer for this suggestion. We have now added this information in line L470.

Around 50 to 60 tubules per sections analyzed per rat.

For sperm counts, indicate whether counts were averaged from both epididymides.

Response: We have now added this information in line L474 - 476.

Total numbers of sperm present in each epididymis were counted after releasing the sperms in 1ml of 1× PBS by mincing the epididymis and the average was determined.

  1. Statistics

You mention Mann–Whitney tests, but for multiple comparisons (e.g., 5d vs 12d vs 60d), a Kruskal Wallis or ANOVA might be more appropriate. Clarify why non-parametric tests were chosen was normality tested?

Response: We thank the reviewer for critical analysis. We did not use Mann–Whitney tests for multiple comparisons in Figure 2 i.e. 5d vs 12d vs 60d Sc in Rat. For the differential expression of NOR1 for Figure 5 A, we used pair t test. The normality was tested for this experiment. However, for the validation of differential expression of NOR1 by q-RT-PCR analysis and fertility analysis (testes weight, sperm count) Mann–Whitney tests were used. The sample size in these experiments was less. Therefore, we did not check the normality, and we used a stronger non parametric test. Moreover, there were two groups in these experiments. So we used Mann–Whitney tests for such study. This is in accordance with our previous studies (PMID: 23667645; PMID: 29064078; PMID: 28065881; PMID: 29363789; PMID: 29032151; PMID: 33145986).  

Discussion

-Strong integration of findings with previous literature.

Response: We thank the reviewer for this suggestion.  We have now rewritten the discussion section.

-Could benefit from highlighting species differences (rat vs. mouse) more explicitly, since you justify using rat models.

Response: We thank the reviewer for this suggestion.  We have added these information in the lines L322-328

Rat models were chosen because our microarray data is from rat and rat are phys-io-logically closer to humans as compared to mice [101, 102]. The mice are widely used for genetic studies due to the advancement in embryo transfer techniques. However, rats have distinct advantages for studying spermatogenesis. In particular, the spermatogenesis re-quires approximately 55days to complete a full cycle in rat which closer to that of human (64 days) [103-105]. Also the rat seminiferous epithelium has 14 defined stages compared to 12 in the mouse [103, 104].

References

  1. Pradhan BS, Majumdar SS. An Efficient Method for Generation of Transgenic Rats Avoiding Embryo Manipulation. . Mol Ther Nucleic Acids. 2016;8;5(3):e293.
  2. Zhao S, Shetty J, Hou L, Delcher A, Zhu B, Osoegawa K, et al. Human, mouse, and rat genome large-scale rearrangements: stability ver sus speciation. Genome Res. 2004;Oct;14(10A):1851-60.
  3. Ahmed EA, de Rooij DG. Staging of mouse seminiferous tubule cross-sections. Methods Mol Biol. 2009;558:263-77.
  4. Clouthier DE, Avarbock MR, Maika SD, Hammer RE, Brinster RL. Rat spermatogenesis in mouse testis. Nature. 1996;30;381(6581):418-21.
  5. El‐Domyati M, Al-Din A, Barakat M, El‐Fakahany H, Xu J, Sakkas D. Deoxyribonucleic acid repair and apoptosis in testicular germ cells of aging fertile men: the role of the poly(adenosine diphosphate-ribosyl )ation pathway. Fertility and Sterility. 2009;91(5):2221-9.

Round 2

Reviewer 2 Report

Comments and Suggestions for Authors

The manuscript is well-written in its current form, and I appreciate the efforts.